# Delocalization of the Unpaired Electron in the Quercetin Radical: Comparison of Experimental ESR Data with DFT Calculations

**DOI:** 10.3390/ijms21062033

**Published:** 2020-03-16

**Authors:** Zhengwen Li, Mohamed Moalin, Ming Zhang, Lily Vervoort, Alex Mommers, Guido R.M.M. Haenen

**Affiliations:** 1Department of Pharmacology and Toxicology, Faculty of Health, Medicine and Life Sciences, Maastricht University, P.O. Box 616, 6200 MD Maastricht, The Netherlands; maxamed.moalin@gmail.com (M.M.); z.ming@maastrichtuniversity.nl (M.Z.); l.vervoort@maastrichtuniversity.nl (L.V.); a.mommers@maastrichtuniversity.nl (A.M.); 2Research Centre Material Sciences, Zuyd University of Applied Sciences, 6419 DJ Heerlen, The Netherlands

**Keywords:** quercetin radical, ESR, unpaired electron, DFT calculation

## Abstract

In the antioxidant activity of quercetin (Q), stabilization of the energy in the quercetin radical (Q^•^) by delocalization of the unpaired electron (UE) in Q^•^ is pivotal. The aim of this study is to further examine the delocalization of the UE in Q^•^, and to elucidate the importance of the functional groups of Q for the stabilization of the UE by combining experimentally obtained spin resonance spectroscopy (ESR) measurements with theoretical density functional theory (DFT) calculations. The ESR spectrum and DFT calculation of Q^•^ and structurally related radicals both suggest that the UE of Q^•^ is mostly delocalized in the B ring and partly on the AC ring. The negatively charged oxygen groups in the B ring (3′ and 4′) of Q^•^ have an electron-donating effect that attract and stabilize the UE in the B ring. Radicals structurally related to Q^•^ indicate that the negatively charged oxygen at 4′ has more of an effect on concentrating the UE in ring B than the negatively charged oxygen at 3′. The DFT calculation showed that an OH group at the 3-position of the AC ring is essential for concentrating the radical on the C2–C3 double bond. All these effects help to explain how the high energy of the UE is captured and a stable Q^•^ is generated, which is pivotal in the antioxidant activity of Q.

## 1. Introduction

In all life forms, opposing forces provide the energy that flows through networks in an organism, which fuels life [1]. In this concept, health is the ability of an organism to maintain the balance between these opposing forces, which creates resilience. The major source of energy in our body is provided by the redox reactions of glycolysis, lipid oxidation, and the citric acid cycle. In a healthy body, the redox energy is high and kept within very strict limits, known as homeostasis [2,3].

Within this concept, the treatment of diseases should focus on adjusting the deranged flow of energy. The deranged energy flow is manifested by the formation of highly reactive oxygen species (ROS) [4]. In the most reactive ROS, the hydroxyl radical, all the energy of its unpaired electron (UE) is located on a single atom (i.e., oxygen). The highly concentrated energy explains why the hydroxyl radical is “hard” and very reactive. It is the most reactive molecule formed in our body. The hydroxyl radical reacts quickly with virtually any molecule it encounters. This will inflict damage to our body and further derange the energy flow. Antioxidants such as quercetin (Q) are hard electron donors that react quickly with ROS that are hard electron acceptors. By accepting electrons, ROS are converted into relatively unreactive compounds, which do not damage the cell. For example, by accepting an electron, the hydroxyl radical is converted into the hydroxyl anion that, after protonation, becomes water [1,5,6]. In the protection against ROS, Q takes over the energy of the ROS and channels this energy in the antioxidant network of the body [7]. When Q takes over the energy, it is firstly oxidized to a quercetin radical (Q^•^). To protect, it is essential that the energy of the UE of in Q^•^ is safely stored in the molecule. It is generally accepted that this is due to the delocalization of the UE in the extensive π-system of Q^•^. Despite its pivotal role in the antioxidant activity of Q, research on the exact delocalization of the UE in Q^•^ is rare and not conclusive [6,8,9,10]. This prompted us to elaborate on this by combing experimentally obtained spin resonance spectroscopy (ESR) results with theoretical quantum calculation of the delocalization of the UE in Q^•^, while also examining some structurally closely related radicals, and comparing our results to previously reported data.

In the ESR experiments, a magnetic field is generated, and the para-magnetic protons in the neighborhood of the UE will give rise to different peaks in the ESR spectrum of an organic radical such as Q^•^. In this way, the ESR spectrum gives information about the delocalization of the UE in the molecule [11]. A disadvantage of the ESR technique is that, of Q^•^, only the spin populations on atoms with protons are obtained. Information about atoms or bonds without a proton nearby (such as oxygen, π-bond conjugated of the aromatic ring) is not obtained, while these ESR silent groups are critical in the delocalization of the UE of Q^•^. Quantum calculations were used to complete the missing part and to verify the experimental ESR results.

## 2. Results

In order to resolve the delocalization of the UE in Q^•^, the ESR spectrum of Q^•^ and that of several structurally related radicals were recorded and analyzed. Density functional theory (DFT) was used for the electronic properties of the compounds. To generate reliable spin population and spin density results of a compound, its three-dimensional (3D) structure has to be accurate. Therefore, the geometry of the radicals in the water phase was determined first. Considering the high pH in the experiments, the fully deprotonated form of the radicals was used. Finally, the ESR data were combined with the DFT calculations.

### 2.1. ESR Spectrums

#### 2.1.1. Hydroquinone and Catechol Radicals

To unravel the complicated ESR spectrum of Q^•^, the ESR spectra of the hydroquinone and catechol radicals were examined first (Figure 1).

Autoxidation of hydroquinone gave a typical quintet [12]. Following Pascal’s triangle, the observed quintet can be explained by the interaction of the UE with four equivalent protons. The hydroquinone radical has four protons that are bound to C2, C3, C5, and C6. The equivalence of these protons is ascertained by the symmetry of the molecule. Their hyperfine coupling constant was found to be 2.337 G.

The ESR spectrum of the catechol radical shows a triplet split into a smaller triplet (Figure 1). The symmetry of the catechol molecule shows that the protons on C3 and C6 are equivalent, and, because of the neighboring electronegative oxygen atom, the small triplet signals with hyperfine coupling of 0.8405 G were assigned to the effect of these protons. The larger triplet with hyperfine coupling of 3.549 G can be assigned to the other two equivalent protons on C4 and C5.

#### 2.1.2. Myricetin, Kaempferol, and Quercetin Radicals

Autoxidation of myricetin (M) led to a distinct triplet ESR signal, indicating the presence of two equivalent protons, with a hyperfine coupling of 1.007 G. The structure of the fully deprotonated myricetin radical (M^•^) shows that four protons may affect the ESR signal, namely, those on C6, C8, C2′, and C6′. Due to the location of a negatively charged oxygen at the 5-position, the protons on C6 and C8 in the A ring are not equivalent and, therefore, would not lead to the observed triplet. Based on the symmetry of the B ring of M^•^, it is most likely that the two equivalent protons that are responsible for the triplet signal are on C2′ and C6′ in the B ring (Figure 2).

The ESR spectrum of the kaempferol radical (K^•^) is a quintet with a hyperfine coupling of 2.522 G. The quintet obtained is equal to that of the hydroquinone radical, although the intensity of the ESR signal obtained of K^•^ is much lower. At first, it was thought that the quintet was caused by the interaction of the UE with four protons on C2′, C3′, C5′ and C6′ in the B ring. This is further addressed in the discussion.

The ESR spectrum of Q^•^ is a doublet, doublet, doublet with hyperfine couplings of 3.184, 1.480, and 0.652 G (Figure 2). This spectrum suggests that there are three different protons with one higher, one medium, and one smaller interaction intensity. Because the three protons are not equal, it is more difficult to assign these protons. The protons on C6 or C8 on the A ring with two negatively charged oxygens on C5 and C7 might interact with the UE. However, considering the results of K^•^ and M^•^ where the protons in the B ring are thought to have the main effect on the ESR signal, the simplest explanation for Q^•^ is that the three protons on the B ring are responsible for the doublets, with the proton on C6′ causing the largest doublet (similar to the catechol radical), the proton on C2′ causing the intermediate doublet, and the proton on C5′ causing the smallest doublet.

#### 2.1.3. Methylated Quercetin Radicals

The three methylated quercetin compounds were synthesized to investigate the influence of the negatively charged oxygen on the delocalization of UE.

Autoxidation of 3MQ gives an ESR signal composed of a large doublet split into a smaller triplet, with hyperfine couplings of 2.884 G and of 0.9385 G, respectively (Figure 3). The triplet signals might be due to the two protons on C2′ and C5′ in the B ring, as these two protons are roughly in the same environment. The proton on C6′ might cause the large doublet signals (similar to catechol radical).

The ESR spectra of 3′MQ^•^ and 4′MQ^•^ are much more complex compared with the spectrum of 3MQ^•^, indicating that the protons on the methoxy group in the B ring affect the delocalization of the UE. The ESR spectrum of 3′MQ^•^, as well as that of 4′MQ^•^, is a triplet, a doublet, and a quartet. Their hyperfine couplings are 2.835, 0.760, and 0.684 G in 3′MQ^•^, and 2.811, 0.7355, and 0.7355 G in 4′MQ^•^ (Figure 3). *O*-methylation of the 3′OH group or of the 4′OH group changes the electronic environment of the protons in the B ring, which makes assigning the hyperfine couplings of 3′MQ^•^ or 4′MQ^•^ to a proton challenging. In order to solve this, we employed DFT calculation.

### 2.2. Structure

An accurate 3D structure of Q^•^ is important for the calculation of its spin population. We firstly optimized the structure of Q at M062X/6-311G (d,p) level. The result of the optimization shows that the dihedral angle (C3–C2–C1′–C2′) formed between the B and C ring is 6.89°. This is close to the X-ray experimental data of 8.0°, indicating that the optimization at this level is accurate. Due to the high pH used in the ESR experiment, the optimization of Q^•^ was carried out on its fully deprotonated form. The optimization shows that the dihedral angle (C3–C2–C1′–C2′) of Q^•^ is 0.072° which means that the molecule is practically planar. The bond length of C2–C1′ is also reduced from 1.46 Å in Q to 1.44 Å in Q^•^. The planar configuration of Q^•^ and the relatively short bond length in Q^•^ indicate that the C2–C1′ single bond between both rings is transformed into a “semi-double bond”, which points toward delocalization of the UE over the B ring and the C ring (Figure 4).

K^•^ and M^•^ were also found to be practically planar, with dihedral angles of 0.02° and 3.58°, respectively. The dihedral angles of 3MQ^•^, 3′MQ, and 4′MQ^•^ are 24.08°, 2.19°, and 11.81°, respectively. The relatively high dihedral angle of 3MQ^•^ probably involves a steric effect of the methoxy group at the 3-position within the B ring. More detailed structural characteristics of investigated compounds are given in the Appendix A (Appendix A).

### 2.3. Spin Density Map and Spin Population

For radicals, the distributions of the alpha electron density and beta electron density are different. The spin density maps of the distribution of the UE of all tested compounds are given in Figure 5.

The spin density map of the hydroquinone radical shows that its UE is delocalized over the whole molecule and concentrated on the oxygen atoms and the carbon bound to the oxygen. The UE of the catechol radical is primarily delocalized around its two adjacent oxygen atoms and the carbon atoms they are bound to. The two equivalent neighboring carbons also have a high UE delocalization. The two other carbons have a lower UE delocalization. The spin density maps of both the hydroquinone radical and the catechol radical confirm the interpretation of their ESR signal given above.

The spin density map of K^•^ shows that its UE is concentrated on C2 and C3, and partly on its B ring. In Q^•^, the UE is less concentrated on the C2 and C3 positions and resides more in the B ring, compared to the UE of K^•^. The difference of M^•^ with K^•^ (i.e., less on the C2 and C3 and more in ring B) is even more pronounced than the difference of Q^•^ with K^•^. This indicates that the negatively charged oxygen in ring B greatly attracts and stabilizes the UE. Another important finding is that methylation of the 3-OH as in 3MQ, drastically reduces the delocalization of UE over the C2–C3 double bond. The methylated group also prevents the planar arrangement of the B ring and C ring, which limits the delocalization over both rings and causes the UE to concentrate in only one ring, which appeared to be the B ring. This is also in line with the relatively high length of the C2–C1′ bond, i.e., the connection of the B ring with the C ring, in 3MQ^•^ compared to Q^•^.

Comparing the spin density map of 4′MQ^•^ to that of Q^•^ shows that methylation of the 4′OH of Q^•^ greatly neutralizes the ability of the negatively charged oxygen in ring B to attract and stabilize the UE in the B ring, and concentrates the UE on the C2–C3 double bond in the C ring. Comparing the spin density map of 3′MQ^•^ and 4′MQ^•^ to that of Q^•^ shows that methylation of the 3′OH has less of an effect than methylation of the 4′OH. This means that, of the negatively charged oxygens in ring B that greatly stabilize the UE, the negatively charged oxygen at the 4′ position has the greatest effect. Interestingly, the spin density map of 3′MQ^•^ closely resembles that of K^•^, which misses the 3′OH group, indicating that methylation of the OH group (which prevents deprotonation and converts the OH group into a negatively charged oxygen) neutralizes the effect of the 3′OH group on the delocalization of the UE.

### 2.4. Combining the ESR Data with the DFT Calculation

By calculating the spin population on the atoms using the splitting constants according to Becke method, the experimental results obtained with the ESR measurements and the DFT calculation were compared. The protons which generate the ESR signal were assigned by (i) ranking the values of the spin density calculation of each proton bound carbon of a specific radical, and (ii) pairing the rank of the calculated spin density to the rank of the value of the hyperfine splitting constant found in the ESR spectrum of that radical (Figure 6).

Although Q^•^ possesses five protons, the experimental ESR results and calculation both indicate that only three of them significantly affect the ESR spectrum. The calculation shows that the spin density of the UE at the C6 and C8 position is too low to affect the ESR signal. Our results suggest that the UE of Q^•^ is delocalized on the B ring, as well as on the C ring. The calculation by the Beck method shows that the absolute spin populations at C2′, C5′, and C6′ are 7.2, 3.7, and 8.0, respectively, indicating that the spin densities calculated using the hyperfine couplings of 5.1, 2.3, and 11 can be assigned to C2′, C5′, and C6′ in the B ring, respectively. The similarity of the value of the experimentally determined spin densities with the spin densities obtained using the DFT calculation supports the accuracy of the DFT calculation. 

The absolute spin population of all four proton-connected carbons of the hydroquinone radical is 6.7, and an experimentally obtained spin density of 8.1 can be assigned to each of them. The geometry of the molecule confirms that these four protons and the carbon atoms they are bound to are equal. The two adjacent negatively charged oxygens of catechol make the C3 and C6 less attractive for UE than C4 and C5. The calculated delocalization for the UE at these sites is 2.7, 2.7, 11, and 11, respectively, which corresponds to the experimental spin densities of 2.9, 2.9, 12, and 12.

M^•^ contains four protons: two protons on its A ring and two protons on its B ring. Calculation shows that the spin density on C6 and C8 of its A ring is relatively low (both 0.2 (Appendix A), suggesting that their adjacent protons are unlikely to affect the ESR spectrum. Calculation also shows that the protons on C2′ and C6′ in the B ring are practically equivalent, with calculated values of 5.0 and 5.2, respectively. The equivalence of these protons is corroborated by the geometry of the molecule. These equivalent protons produce the triplet in the ESR signal, which has the experimental spin density of 3.5.

The calculation of 3MQ^•^ shows that the proton on C8 in the A ring does not affect the ESR spectrum, which is also the case in Q^•^ and M^•^. The protons on C2′ and C5′ in the B ring of 3MQ^•^ are practically equivalent with values of 3.1 and 3.0, respectively, giving the triplet in the ESR spectrum. The calculated spin density on C6′ is relatively high, with a value of 7.8, which gives the doublet.

The spin density maps show that the UEs in Q^•^ and M^•^ are less “soft” than the UEs in 3′MQ^•^ and 4′MQ^•^, as C8 of the A ring is also favored by the UE in 3′MQ^•^ and 4′MQ^•^. The calculated spin densities on C6′ and C8 in 3′MQ^•^, as well as in 4′MQ^•^, are higher than any other proton-connected carbon in these molecules; at C6′, the values are 6.3 and 10 (for 3′MQ^•^ and 4′MQ^•^, respectively) and, on C8, the values are 5.6 and 6.6 (for 3′MQ^•^ and 4′MQ^•^, respectively). This also indicates that the protons at C6′ and C8 in 3′MQ^•^, as well as those in 4′MQ^•^, are practically equivalent and, therefore, generate a triplet in their ESR spectrums. The proton on C5′ in 3′MQ^•^ and the proton on C2′ in 4′MQ^•^ in their B rings generate a doublet. Furthermore, with the three types of methylated quercetin radicals, the calculated spin densities are in line with the experimental spin densities.

The calculation shows that the delocalization of the UE on the proton-connected carbons of the B ring of K^•^ are 4.3 and 4.1 on C2′ and C6′, and 1.3 and on C3′ and C5′, indicating the presence of two pairs of practically equivalent protons in ring B (as corroborated by the geometry of the molecule). The UE of K^•^ is also highly concentrated on C8 of its A ring with a calculated spin density of 5.8. The calculated spin distribution of the UE of K^•^ does not correspond to the ESR signal we obtained experimentally via the autoxidation of K. In several other studies [13,14], more complex spectra than the spectrum we obtained were found after the autoxidation of K. This makes analyzing the delocalization of UE of K^•^ puzzling. This is further elaborated in the discussion.

## 3. Discussion

The principle of the protective effect of a free radical scavenger antioxidant is that (i) the antioxidant quickly takes over the high energy contained in a reactive free radical, (ii) the energy is stabilized by forming a relatively un-reactive antioxidant radical, and (iii) the energy is safely channeled in the antioxidant network of the cell [10]. Q is one of the most studied free radical scavengers because of its extraordinary high potency and, therefore, we focused on this compound [7,15]. In the first step, Q donates an electron to the free radical through hydrogen atom transfer (HAT), sequential proton-loss electron transfer (SET-PT), or single electron transfer followed by proton loss (SPLET), depending on the solvent. In a HAT mechanism, the proton and electron are transferred in the same kinetic process, while in SET-PT or SPLET mechanisms, the electron transfer and proton transfer are two independent kinetic processes; the difference between these two is the sequence of the kinetic processes [16,17]. The HAT mechanism is more plausible for scavenging of radicals by Q in acetone and ethanol [18], while, in aqueous solution, the SPLET mechanism is thermodynamically preferred [19]. The formed Q^•^ is relatively stable due to the delocalization of its UE. Although it is generally accepted that the delocalization is pivotal in the antioxidant activity of Q, the exact delocalization of the UE in Q^•^ is not well determined. The aim of this paper is to further examine the delocalization of the UE in Q^•^, and to elucidate the importance of the A, B, and C ring for the stabilization of the radical by combining experimental ESR measurements with theoretical DFT calculations.

The ESR signal of Q^•^ has eight peaks, composed of one large doublet, one medium doublet, and one small doublet. This splitting of the spectrum of Q^•^ is of similar to that reported by Kuwabara et al. [20], who oxidized Q to Q^•^ in an Na_2_CO_3_^−^/NaHCO_3_ (0.1 M) buffer (pH 10) under nitrogen atmosphere using the ^15^N-labeled sodium salt of nitrosodisulfonate. The proton hyperfine coupling constants they obtained were 1.36, 0.76, and 2.56, assigned to the protons on C2′, C5′, and C6′ in the B ring, respectively. The previously reported ESR spectrum obtained by autoxidation of Q in dimethyl sulfoxide (DMSO)–H_2_O–KOH showed similar hyperfine coupling constants of 1.45, 0.70, and 2.70 that were also assigned to the same protons [12]. These hyperfine coupling constants are in the same range as those we obtained, i.e., 1.480, 0.652, and 3.184. This indicates that Q^•^ was successfully generated. The small differences in the obtained values probably originate from the difference in the oxidation procedures and solvents used. Assigning the protons appropriately relied on chemical intuition and was sometimes incorrect. For example, in some reports, the lowest hyperfine constant was assigned to the proton on C2′ of Q^•^. However, our DFT calculation indicates that this has to be the proton on C5′, which is in agreement with the report by Kuwabara et al. [20]. Q at the high pH we used is fully deprotonated, indicating that Q cannot donate a hydrogen atom, implying SPLET as the dominant mechanism for generating Q^•^.

Spin population results of Q^•^ show that delocalization of the UE on C2′, C6′, and C5′ was 7.2, 8, and 3.7, respectively. This is in line with the experimental data obtained using ESR. The calculated value at C5′ is a bit higher, and that on C6′ is a bit lower compared with the experimental data. This might partly be attributed to a solvent effect, since the experiments were carried out in a mixed DMSO–water solution while the calculation was in water. Actually, the solvent might not only affect the splitting intensities but also affect the distribution of UE. Calculation shows that, in Q^•^, with only one deprotonated hydroxyl group in the gas phase, the UE of Q^•^ is largely localized on the negative charged oxygen [21], while, in a polar solvent, the UE is not concentrated on the negatively charged oxygen [22]. Our calculations also show that the delocalization of the UE in the gas phase differs from that in water (data not shown). The calculation also shows that the total delocalization on the negatively charged oxygen (i.e., on the charged oxygen at the 3′-, 4′-, 3-, 5-, and 7-position with a delocalization of 8.7, 16.1, 8.7, 0.7, and 0.7, respectively) is only 34.9%. Instead, the UE of Q^•^ is mainly delocalized on C2–C3 and on the carbons in the B ring.

Another interesting finding is that, upon increasing the number of negatively charged oxygens on the B ring, more of the UE of the flavonoid radical is delocalized on the B ring. This is as expected as it is generally believed that the oxygen can donate parts of its electrons and, thus, stabilize an UE around it. This explains an enrichment of UE going from K^•^ to Q^•^ to M^•^, which contain respectively one, two, and three deprotonated hydroxyl groups in their B ring. As a consequence, Q^•^ and M^•^ are more likely to donate a second electron compared with K^•^, which corresponds to the ranking of the antioxidant potency of these three flavonoids [18,23].

The spin density map of Q^•^ shows that the beta electron delocalizes on C2 in the C ring, whereas the alpha electron delocalizes on C1′ in the B ring. This suggests an electronic interaction between both carbons on the connection between the B ring and the C ring and that, in an alkaline environment, a semi-double bond is formed between these two rings. Moreover, (i) the reduction in the dihedral angle between the B ring and the C ring, from 6.89° of Q to 0.14° of fully deprotonated Q^•^, and (ii) the reduction of the C2–C1′ bond length from 1.46 Å to 1.44 Å are both consistent with the formation of the semi-double bond between the B ring and the A ring, and with the UE electron being delocalized not only in the B ring, but also in the AC ring of Q^•^.

Comparing the spin density maps of 3MQ^•^ with Q^•^, 3′MQ^•^, and 4′MQ^•^ indicates that the negatively charged oxygen at the 3-position in ring C plays a pivotal role in the distribution of UE. Once the 3-OH group is *O*-methylated, the UE is mainly localized on the B ring. This involves a steric effect of the methyl group, which forces the B ring out of the plane of the AC ring, preventing the delocalization of the UE over both rings since their π-systems are not well connected. Another important reason is that *O*-methylation of the 3-OH group prevents it from deprotonation, whereby no negatively charged oxygen is formed and no “spare electron” is available to stabilize the UE at that position.

Comparing the spin maps of the methylated Q derivatives also indicates that the 4′OH group is more critical for the attraction and delocalization of UE in the B ring than the 3′OH group. Previous experiments revealed that the antioxidant potency (2,2′-azino-bis(3-ethylbenzothiazoline-6-sulfonic acid) (ABTS) scavenging ability) of 4′ methylated Q derivatives is less than that of 3′ methylated Q derivatives [24]. This confirms that a higher concentration of the UE in the B ring is associated with a higher antioxidant activity.

The interpretation of the ESR spectrum of K^•^ is challenging because the experimental data do not correspond to the calculated data. The ESR spectrum obtained after incubating K at a high pH is similar with to that of Kuhnle et al. [12] and Pirker et al. [13] and consists of a quintet. Our first interpretation was that the quintet might be caused by the interaction of the UE with four equivalent protons on C2′, C3′, C5′, and C6′ in the B ring. However, the DFT calculation of K^•^ shows that the spin populations of these carbons on the B ring are not equal. Moreover, the DFT calculation also shows that the C8 of K^•^ causes a significantly higher hyperfine splitting in the ESR spectrum than the protons in the B ring, because C8 has a spin population with a relatively high value of 5.8. The simulated ESR spectrum based on our DFT calculation (Appendix A) resembles the ESR spectrum found experimentally by Pirker et al. [13] after incubating K in an alkaline solution under an argon atmosphere for 5 min. Pirker et al. [13] and van Acker et al. [14] hypothesized that, in an alkaline solution, part of K decomposes, leading to the rapid formation of hydroquinone, and that the quintet ESR signal is that of the hydroquinone radical and not of K^•^. As our results also show, the ESR spectra we obtained with K and hydroquinone are similar. Analysis of a solution of K incubated for 5 min at high pH, similar to the incubation of K in the ESR experiment, confirmed the formation of hydroquinone (Appendix A), substantiating the explanation that the ESR signal with K was derived from the hydroquinone radical.

The intensity of the ESR spectrum we obtained via the autoxidation of K is weaker than that of hydroquinone, which can be explained by the relatively low amount of hydroquinone formed from K during the relatively short time between preparing the reaction mixture and recording the ESR spectrum (Appendix A). Moreover, the difference between our simulated ESR spectrum of K^•^ and the ESR spectrum of K^•^ experimentally found by Pirker et al. [13] might be due to formation of some hydroquinone radicals in their incubation. For a better comparison, the simulated spectrums of other tested compounds were drawn (Appendix A). 

This spin distribution of K^•^ obtained with the DFT calculation is consistent with the spin distributions of the other compounds discussed in the present study for which the experimental data did correspond with the calculated data. K has fewer OH groups its B ring compared to Q and M, which explains why relatively less of the UE in K^•^ is attracted to its B ring, while more resides in its AC ring. This is consistent with the result that the dihedral angle of K^•^ is even smaller than that of Q^•^ and M^•^, and that the bond length of C2–C1′ is the shortest among these three compounds with a length of 1.42. This is also consistent with the result that the HOMO(SOMO)-LUMO gap of K• is the highest comparing with Q• and M•, indicated that the K• is less reactive and might be more stable (Appendix A).

In addition to the tested compounds, we also tried to record the ESR spectrum of galangin, a compound that has the same AC ring as K, M, and Q, but with no hydroxyl group in its B ring. We failed to generate an ESR spectrum with galangin under the same experimental conditions we used for the other compounds (data not shown). Galangin is a potent free radical scavenging antioxidant, and we expected that a galangin radical would have been formed under the conditions we used. A possible explanation for failing to detect a galangin radical is that the half-life of the galangin radical is even shorter than that of K^•^; thus, so the galangin radical is degraded before it can be detected in our experiments. This would be in line with our conclusion that hydroxyl groups in ring B play a key role in stabilizing the UE, as galangin lacks a hydroxyl group in the B ring. The different behavior of galangin would also corroborate the presence of two antioxidant pharmacophore moieties in flavonols as suggested previously, namely, the AC ring and the B ring [25], which seem to have different properties. We hypothesize that, in the antioxidant activity of Q, the AC ring pharmacophore in Q donates an electron to a radical, and, as indicated in the present study, the UE of the formed Q^•^ is safely stored the B ring pharmacophore.

## 4. Materials and Methods

### 4.1. Chemicals

Quercetin∙2H_2_O was purchased from Acros Organics (Geel, Belgium) and used as received. Hydroquinone, kaempferol, and myricetin were obtained from Fluka (Buchs, Switzerland). Catechol was purchased from Janssen Chimica (Beerse, Belgium). All chemicals were at least of 95% purity. The methylated derivatives of quercetin were synthesized as described previously [24]. The structures of the tested compounds are given in Figure 7.

### 4.2. Autoxidation and ESR Procedures

Autoxidation was performed as described by Pirker et al. [13]. Briefly, the compounds were dissolved in DMSO to a concentration of 1–5 mM. Then, 650 μL of the DMSO solution was added into an ESR flat cell, and 250 μL of a 0.1–1 M sodium hydroxide solution was added. The flat cell was immediately placed in the ESR spectrometer (Bruker BioSpin, Rheinstetten, Germany). The recordings were at 10 kHz of modulation frequency and 1 G of amplitude on a Bruker ESP300 ESR spectrometer. The microwave frequency was set at 9.79 GHz, with a power of 20 mW. Magnetic field sweep widths were in the range 0.9–2.0 mT, depending on the spectral widths. A modulation amplitude of 0.01 mT was used for most measurements. The spectra given are the sum of 10 successive scans of the same sample that were all recorded in 5 min. The spin populations (ρπ) of the proton-bound carbon atoms were calculated by O’Connell’s equation, a = 29*ρπ, from the hyperfine couplings obtained [26].

### 4.3. Calculation Details

The equilibrium geometries of all compounds (radical forms) were fully optimized with Gaussian 09 package [27] using the DFT method at the M062X [28]/6-311G (d,p) [29] level. In order to calculate the spin populations of investigated compounds, their energies were obtained at the PWPB95 [30]/ma-def2-TZVPP [31] level by orca [32], whereas Grimme’s DFT-D3 dispersion correction was also employed [33]. The solvent effects on the tested compounds were taken into account by application of implemented Solvation Model Density (SMD, Water) method [34].

The spin density maps were generated with the help of Multiwfn [35] and VMD [36]. Multiwfn was also applied for the calculation of spin population with Becke methods.

### 4.4. Statistics

All ESR experiments were performed at least in triplicate. Data are expressed as means ± SD or as a typical example.

## 5. Conclusions

In conclusion, our experimental ESR measurements and theoretical DFT calculations appear to give similar results that are in line with previous reports, which makes the outcome trustworthy. The UE of Q^•^ is mostly delocalized in the B ring and partly on the AC ring. The negatively charged oxygen group at the 4′ position and, to a lesser extent, the negatively charged oxygen group at the 3′ position have an electron-donating effect that stabilizes the electron-deficient π system in the B ring. In addition, an OH group at the 3-position of the AC ring is essential for concentrating the radical on the C2–C3 double bond. The delocalization in the C2–C3 double bound, especially in the B ring, stabilizes the UE in Q^•^ such that the energy it contains is relatively safely stored.

## Figures and Tables

**Figure 1 ijms-21-02033-f001:**
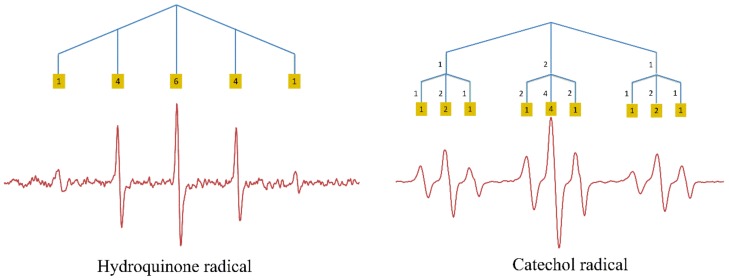
Experimentally obtained spin resonance spectroscopy (ESR) spectra of the hydroquinone radical and the catechol radical with the protons causing the splitting of the signal. The relative intensities of the peaks are given in yellow squares.

**Figure 2 ijms-21-02033-f002:**
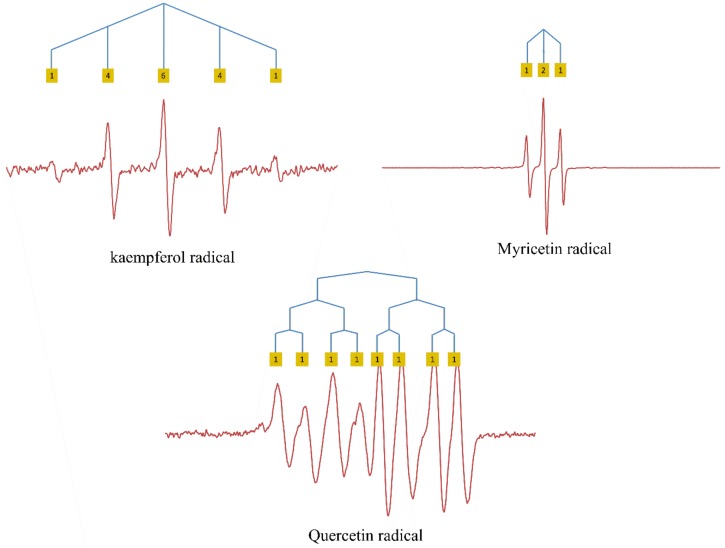
ESR spectra of flavonol radicals with the protons causing hyperfine couplings indicated. The relative intensities of peaks are given in yellow squares.

**Figure 3 ijms-21-02033-f003:**
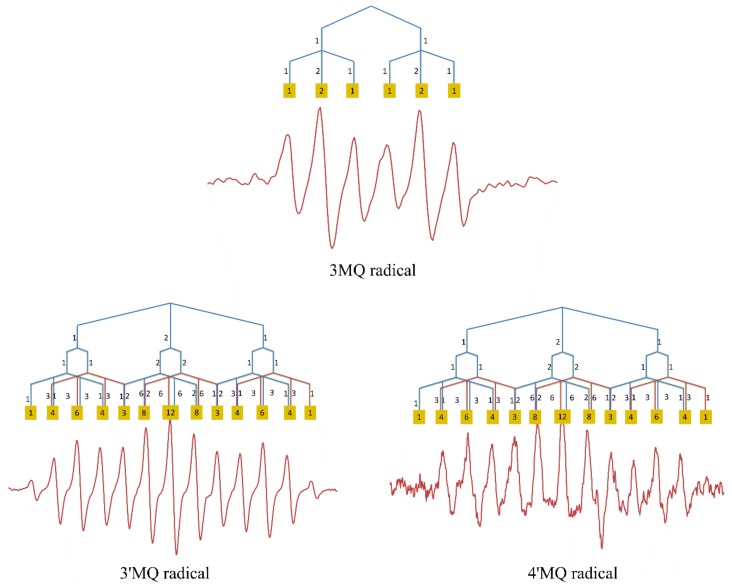
ESR spectra of the radical of the *O*-methylated derivatives of Q, with the protons causing hyperfine couplings indicated. The relative intensities of peaks are given in yellow squares.

**Figure 4 ijms-21-02033-f004:**
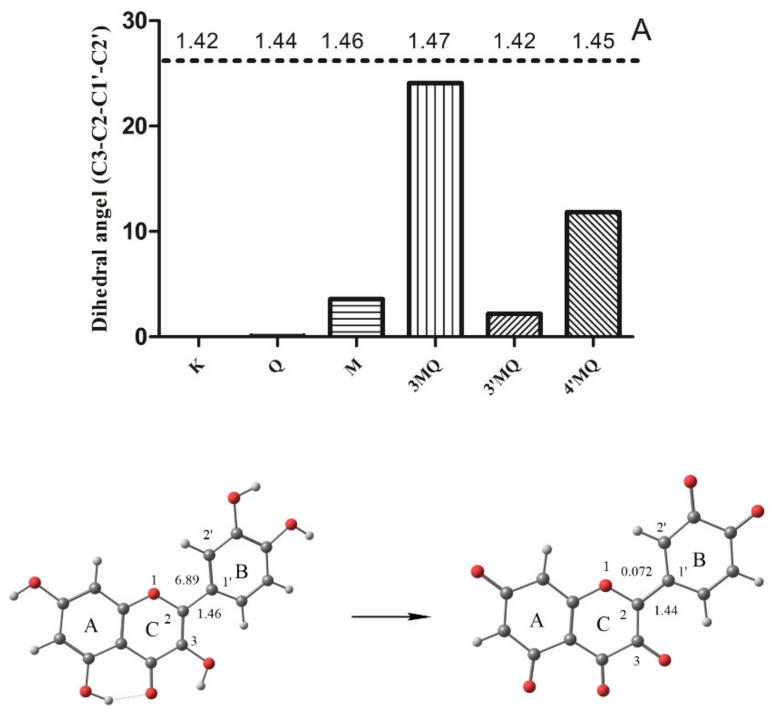
The dihedral angle (C3–C2–C1′–C2′) of tested compounds and the optimized structure of Q and fully deprotonated Q^•^ in the water phase. The dihedral angle (C3–C2–C1′–C2′) of 0.072° indicates that Q^•^ is planar; A: bond length.

**Figure 5 ijms-21-02033-f005:**
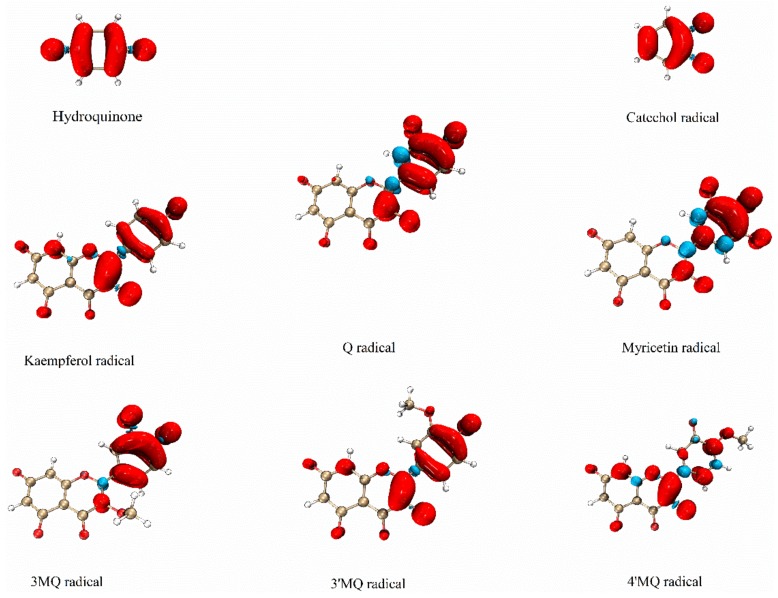
The spin density map of the deprotonated radicals in water. The red color represents the alpha electron and the blue color represents the beta electron.

**Figure 6 ijms-21-02033-f006:**
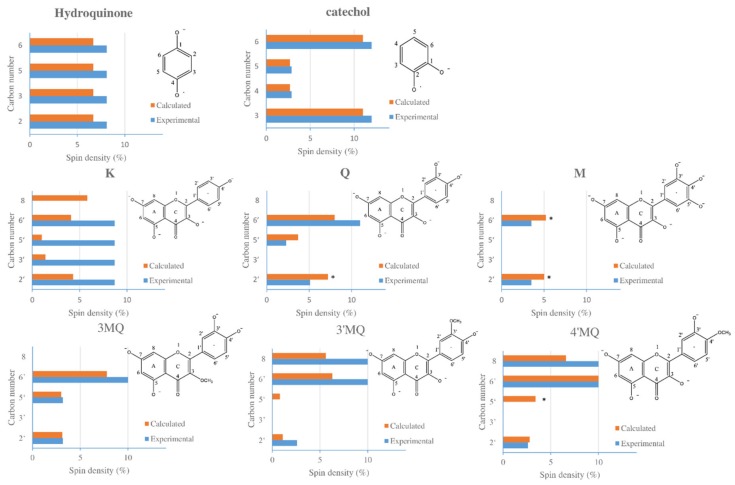
The experimental and calculated spin populations of proton-bound carbons of tested compounds. * representing β electrons (the values of the β electron are negative; for a better comparison, we used absolute values in the picture above, noting that the calculated spin populations are in percentages. This also applies to the whole text when talking about calculated data).

**Figure 7 ijms-21-02033-f007:**
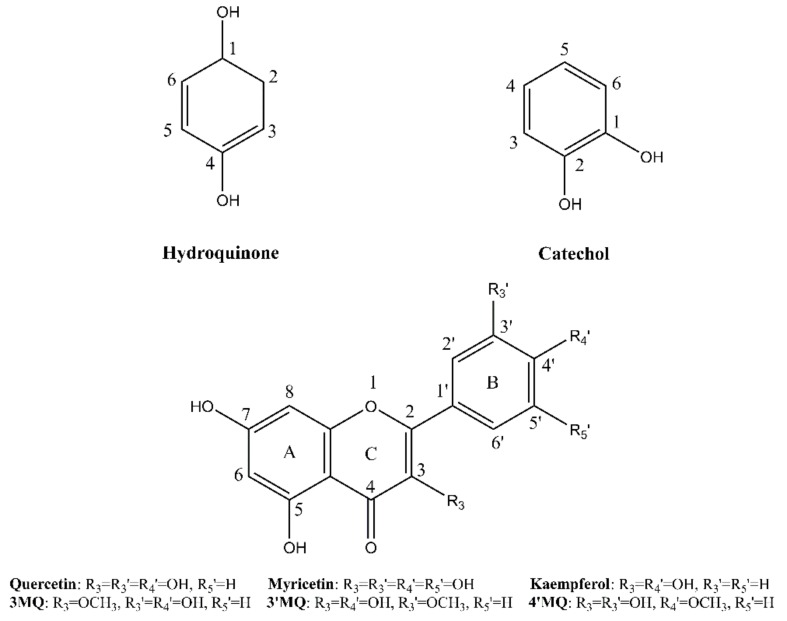
The structures of the tested compounds.

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
