# Peer review of "Delocalization of the Unpaired Electron in the Quercetin Radical: Comparison of Experimental ESR Data with DFT Calculations"

_ijms, 2020, doi:10.3390/ijms21062033_

Round 1
Reviewer 1 Report
The potential health benefits of the antioxidant and anti-inflammatory activities of quercetin have been widely studied and clarified based on the molecular, cellular and structure activity relationships. However, the studies on the anticancer properties of quercetin mostly focused only on the molecular and cellular mechanisms. This interested manuscript of Li et al., introduced new insights for the stability of quercetin radical that may resolve the mystery of the correlations between anticancer activity quercetin and its chemical structure that has not been studied yet. Generation of the stable quercetin radical is due to the delocalization of the unpaired electron that safely stored in the C2-C3 double bound in the B-ring pharmacophore of quercetin.
Author Response
We would like to thank reviewer 1 for the time spent in reviewing our manuscript and for the constructive review report. Based on the review report, we were pleased to note that the message of our manuscript was clear to the reviewer. Since no additional remarks for changing the manuscript were made by the reviewer, no changes based on the review report were made.
Reviewer 2 Report
The paper describes the delocalization of the UE of Q to better understand it stabilization energy and its antioxidant activity. Comparison of EPR and DFT calculations have been performed. The work is clear but sometimes too much basic information is given taking into account this is a research paper to describe novel information. In this regard, there is a nice explanation about ESR for readers that are not experts but maybe too much. This section could be summarized adding some references. In Section 2.1.1. all information is well known and could be also summarized or even removed adding a reference. Only the key information to understand the new findings of the paper should be written. In fact, EPR of Q is known but it is very nice how this paper correlates the spectra with the delocalization of the UE along the molecule and the use of calculations to corroborate the EPR experimental findings. More aspects that the authors must take into account are:
- The first time UE appear in the abstract should be written the long name of it.
- Important information about the use of Quercetin is described in this reference and it’d be nice to be cited: DOI:10.1038/s41598-017-12072-5
- Draw what are B and AC rings in the schemes
- It’d be nice if the experimental EPR spectra are compared with simulated ones using a software (i.e. with EasySpin or the ones of Brucker).
- The theoretical part should be revised by theoreticians and if it is possible the energy of the molecular orbitals calculated to further understand the stabilization/energy of the radical.
- We know it is not the aim of the paper but as organic radicals are electroactive molecules, it would be also very nice if the authors could add the cyclic voltammetry spectra for this interesting radicals and compare the energy with the potential of oxidation/reduction for each one
- For non-experts HAT, SET-PT, SPLET, are not clearly stated. If the authors need to mention them, they need to be explained a little bit more. At least, explain the main differences between these mechanisms.
- 4.2. the title need to include also “Calculation details”
Author Response
We would like to thank reviewer 2 for the time spent in reviewing our manuscript and for the constructive review report. Based on the review report, we were pleased to note that the message of our manuscript was also clear to reviewer 2. Our response to Comments and Suggestion made by this reviewer are:
- The work is clear but sometimes too much basic information is given taking into account this is a research paper to describe novel information. In this regard, there is a nice explanation about ESR for readers that are not experts but maybe too much. This section could be summarized adding some references. In Section 2.1.1. all information is well known and could be also summarized or even removed adding a reference. Only the key information to understand the new findings of the paper should be written.
The paper is not only intended for researchers doing ESR/EPR research, and therefore we included a concise introduction to ESR. Based on the comment of the reviewer, we carefully examined the information given on ESR in the introduction of the manuscript. This basic information discusses the limitation of the ESR and explains why we combine the experimental ESR data with DFT calculation. As requested by the reviewer, we added a ref [11] line 56, for more theoretical information. In section 2.12, the ESR spectrum of the hydroquinone radical and catechol radical are given. Although these ESR spectrum are well known, as reviewer kindly pointed out, that are very helpful to demonstrate the ESR spectrum of the kaempferol radical and the other radicals. We hope that after this explanation the reviewer can agree with keeping this information.
- The first time UE appears in the abstract should be written the long name of it.
This has been corrected. Line 14.
- Important information about the use of Quercetin is described in this reference and it’d be nice to be cited: DOI:10.1038/s41598-017-12072-5
The reference proposed by the reviewer has been added. See line 45 and line 244, ref [7].
- Draw what are B and AC rings in the schemes
The B and AC rings have been drawn in the schemes of the revised manuscript. In the revised Figure 4 and abstract schemes, the B and AC rings are drawn. Figure 7, also have shown where the B and AC ring are(chemical section, due to the request of Journal, Figure 7 has been put at the end of the manuscript)
- It’d be nice if the experimental EPR spectra are compared with simulated ones using a software (i.e. with EasySpin or the ones of Brucker).
The simulated ESR/EPR spectra are given in the supplemental data. See SI 5.
- The theoretical part should be revised by theoreticians and if it is possible the energy of the molecular orbitals calculated to further understand the stabilization/energy of the radical.
The aim of our study was to examine how the unpaired electron is delocalized in the radicals. The energy of the molecular orbitals is indeed interesting information, but it will not give direct information on the delocalization of the unpaired electron. However, as requested, we put the HOMO(SOMO)-LUMO gap map in the supplementary section. See SI 4.
SI 4. The HOMO(SOMO) and LUMO gap of the tested compounds (in the radical form). The gap between alpha-HOMO(SOMO) and alpha-LUMO of kaempferol radical, quercetin radical and myricetin radical are 5.15, 5.10, 5.04 eV, respectively, which indicates that the kaempferol radical is more stable than the quercetin radical and the myricetin radical. The higher alpha energy gap of 4’MQ radical compared with 3’MQ radical indicates the importance of hydroxyl group in Q at the 4’ position. The default isosurface is 0.05 that’s why the alpha-LUMO of catechol radical and hydroquinone radical can’t be shown. The SOMO maps of tested compounds after orthogonal normalization are shown on the right.
It also should be pointed out, as the calculation method is under the unrestricted open shell, we had two individual orbital systems, alpha, and beta. Often alpha is of main interest.
We presented the calculation details in an individual section of our manuscript. (section 4.3: Calculation details).
- We know it is not the aim of the paper but as organic radicals are electroactive molecules, it would be also very nice if the authors could add the cyclic voltammetry spectra for this interesting radicals and compare the energy with the potential of oxidation/reduction for each one
As also acknowledged by the reviewer, a cyclic voltammogram is indeed interesting information, but it will not give direct information on the delocalization of the unpaired electron. Also, because we do not have the possibility to do cyclic voltammetry, we have not included it in our manuscript. We hope the reviewer can agree with this.
- For non-experts HAT, SET-PT, SPLET, are not clearly stated. If the authors need to mention them, they need to be explained a little bit more. At least, explain the main differences between these mechanisms.
We have given the requested explanation, see line 245-250, we also added ref [16] and [17].
……either through hydrogen atom transfer (HAT), or sequential proton-loss electron transfer (SET-PT), or single electron transfer followed by proton loss (SPLET), depending on the solvent. In a HAT mechanism, the proton and electron are transferred in the same kinetic process. In SET-PT or SPLET mechanism, the electron transfer and proton transfer are two independent kinetic processes, the difference between these two is the sequence of the kinetic processes [16, 17]……
- 2. the title need to include also “Calculation details”
Based on the comment of the reviewer we changed the title to: Delocalization of the unpaired electron in the Quercetin radical: Comparison of experimental ESR data with DFT calculation.